# LESIONS, LATENTS, AND LANGUAGE: INTERPRETING BREAST ULTRASOUND FEATURES VIA LATENT PROBING AND LLM-DRIVEN REPORT SYNTHESIS

## ABSTRACT

Interpretability remains a critical bottleneck in the deployment of deep learning models in medical imaging. In this work, we present a novel framework that bridges deep visual representations with natural language through the lens of breast ultrasound lesion segmentation. Using the BUS-UCLM dataset, we train a supervised segmentation model and extract high-dimensional latent features that encode lesion characteristics. We then project these features into a low-dimensional latent space via t-SNE and identify visually coherent clusters. Each cluster is quantitatively characterized using lesion-level attributes such as size, boundary complexity, and class prevalence. To close the semantic gap between neural representations and clinical reasoning, we prompt large language models (LLMs) with these cluster-level summaries to generate human-interpretable natural-language descriptions of lesion types and patterns. Our experiments demonstrate that these language outputs align well with known clinical lesion types and that probing classifiers trained on latent features alone achieve strong diagnostic separation. This framework enables transparent, cluster-driven summarization of lesion types and offers an explainability interface between deep neural models and clinical end-users. Our results suggest a new path for integrating unsupervised vision-language synthesis into medical imaging pipelines without the need for textual ground-truth reports.

## 1 INTRODUCTION AND BACKGROUND

Deep learning has achieved remarkable success in medical image segmentation, with convolutional neural networks (CNNs) and their variants consistently delivering state-of-the-art results across modalities (Xu et al., 2019). In breast ultrasound (BUS) imaging, accurate lesion segmentation can support early cancer detection, yet the opaque nature of these models limits their clinical adoption. Most high-performing architectures, including U-Net derivatives, SegNet(Badrinarayanan et al., 2017) , and DeepLabV3+(Chen et al., 2018b) , operate as black boxes, producing segmentation masks without conveying the reasoning behind their predictions (Gómez-Flores & Pereira, 2020). Despite achieving F1-scores $> 0.90$ and IoU $> 0.81$, interpretability remains a critical barrier, particularly in BUS, where lesion appearance is highly variable due to speckle noise, low contrast, and patient-specific anatomy (Amorim et al., 2021; Ansari et al., 2024).

Interpretability strategies such as saliency maps, Grad-CAM, and attribution methods are widely adopted for highlighting influential regions in medical images (Huff et al., 2021). However, their effectiveness is limited. For example, Saporta et al. showed that Grad-CAM performs poorly in capturing diagnostically relevant regions in chest X-rays (Saporta et al., 2021), while comparative studies of Grad-CAM, LIME, and SHAP highlight varying strengths and weaknesses across pathologies (Barra et al., 2024). Brima & Atemkeng (Brima & Atemkeng, 2024) further demonstrated that ScoreCAM and XRAI improve region retention, yet these methods often fail to provide structured lesion-level insights aligned with radiological reasoning. This underscores the broader challenge of conveying high-level semantics and ensuring robustness across diverse imaging modalities (Amorim et al., 2021).

In parallel, unsupervised clustering of latent representations has emerged as a promising direction for discovering clinically meaningful patterns without reliance on labels. Clustering approaches have been successfully applied to medical concepts from electronic health records (Jaume-Santero et al., 2022), to image patches with deep embeddings (Perkonigg et al., 2020), and to features extracted from CT scans to identify radiologically relevant groups (Hofmanninger et al., 2016). Dimensionality-reduction techniques such as t-SNE and UMAP reveal non-linear relationships in embeddings (Jaume-Santero et al., 2022), while Chen (Chen, 2023) proposed anchor-based visualization for neural survival analysis models. These methods highlight the potential of unsupervised feature discovery, yet they rarely connect such findings to natural-language outputs interpretable by clinicians.

Recent advancements in medical vision-language models (Med-VLMs) attempt to bridge the gap between visual data and clinical reasoning. Transformer-based multimodal architectures have been applied for report generation, visual question answering, and diagnosis (Chen et al., 2024). While they demonstrate strong capabilities, most approaches rely on paired image–text datasets (e.g., radiology reports linked to scans) that are scarce, heterogeneous, and often restricted due to privacy concerns (Hartsock & Rasool, 2024). Vision-language pretraining (VLP) strategies, combining paired and unpaired datasets through self-supervised learning, offer a partial remedy (Shrestha et al., 2023). Nonetheless, key challenges remain: limited ultrasound-specific text corpora, inconsistent evaluation metrics, and the need for interpretability and robust regulatory frameworks (Kalpelbe et al., 2025).

Taken together, prior work has advanced BUS segmentation, interpretability techniques, unsupervised clustering, and multimodal integration. Yet, there is still no unified framework that connects latent visual representations from top-performing segmentation models to clinically meaningful natural-language explanations without requiring paired image–text data. To address this gap, we introduce a segmentation-driven latent probing framework for BUS interpretation. Our approach combines high-performance lesion segmentation, unsupervised clustering of latent features, quantitative cluster characterization, and large language model (LLM)-based report synthesis. By extracting lesion embeddings, discovering visually coherent clusters, and prompting LLMs with cluster-level quantitative attributes (e.g., lesion size, boundary complexity, class distribution), our pipeline generates transparent, human-interpretable lesion descriptions. This enables a novel pathway for explainable AI in BUS diagnostics, bridging the semantic divide between black-box segmentation models and clinical reasoning.

## 2 MATERIALS AND METHODS

We used the Breast Ultrasound (BUS-UCLM) dataset (Vallez et al., 2025) for all training and evaluation. The dataset consists of 683 ultrasound images from 38 patients, acquired between 2022 and 2023 using a Siemens ACUSON S2000 system. Each image is accompanied by an expert-annotated segmentation mask, with red, green, and black regions representing malignant, benign, and normal tissue, respectively. The images span three categories: 419 normal, 174 benign, and 90 malignant. To ensure consistency and manage computational requirements, all images and their corresponding masks were resized to a fixed resolution of $256\times256$ pixels before being fed into the models.

To establish a robust foundation for subsequent latent space analysis, we implemented a set of widely used deep learning architectures for medical image segmentation. These models were selected to represent diverse architectural strategies for capturing contextual information and refining lesion boundaries. As a baseline, we include the original U-Net encoder–decoder with skip connections (Ronneberger et al., 2015). Variants of U-Net incorporating pre-trained ResNet backbones and attention gates were also implemented (Zhou et al., 2018; Oktay et al., 2018). Beyond this family, we evaluated SK-UNet, which integrates Selective Kernel (SK) convolutions(Li et al., 2019) to adaptively capture multi-scale features (Li et al., 2019), and DeepLabv3(Chen et al., 2018a), which employs Atrous Spatial Pyramid Pooling (ASPP) to aggregate contextual information across multiple receptive fields (Chen et al., 2018a). Additional models included RefineNet(Lin et al., 2017), which performs high-resolution segmentation through multi-path refinement (Lin et al., 2017), and SegNet(Badrinarayanan et al., 2017), an encoder–decoder network that leverages pooling indices for efficient upsampling (Badrinarayanan et al., 2017).

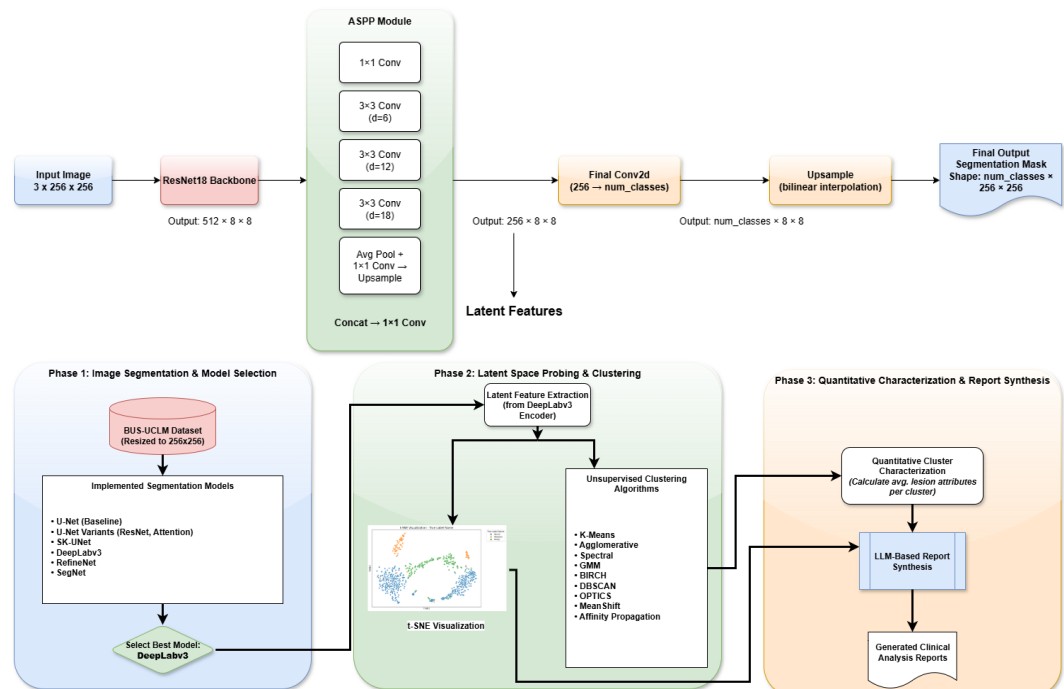

Figure 1: DeeplabV3(Chen et al., 2018a) Model and System Pipeline Overview

For Segmentation Model training, we employed a cosine annealing learning rate scheduler with an initial learning rate of $1 \times 10^{-4}$, $T_{\max} = 100$, and a minimum learning rate of $1 \times 10^{-6}$. Early stopping was applied with a patience criterion based on validation loss, using a stopping threshold of $1 \times 10^{-4}$ for four consecutive epochs. Each model was trained until training was stopped if the early stopping condition was met.

All models in our study were trained under identical experimental conditions to ensure fairness and reproducibility. We adopted a 5-fold cross-validation strategy, which not only provided a more reliable estimate of generalization performance but also reduced the risk of overfitting to a particular train–test split. Training for each fold was continued until convergence, thereby guaranteeing stability in the obtained results and preventing premature termination that might bias performance comparisons.

For evaluation, we employed a comprehensive set of quantitative metrics widely used in medical image segmentation. These included Intersection-over-Union (IoU) and the Dice coefficient, both of which capture spatial overlap between predicted masks and ground-truth annotations; pixel-wise accuracy, which reflects the overall correctness of predictions; and precision, recall, and F1 score, which provide deeper insights into class-specific detection reliability. Together, these metrics allowed us to capture both global and lesion-level performance trends.

To investigate internal representations of the segmentation network, we extracted high-dimensional feature vectors from the encoder of the DeepLabv3(Chen et al., 2018a) architecture (Chen et al., 2018a). For each lesion region, the corresponding latent features were collected to serve as the basis for downstream interpretability analyses. To enable qualitative inspection of the latent space, we applied t-SNE (van der Maaten & Hinton, 2008) to project the high-dimensional feature vectors into a two-dimensional plane. This step facilitated visualization of how lesion samples were organized in the learned embedding space. In addition, we evaluated multiple unsupervised clustering algorithms on the extracted latent features, including k-means(MacQueen, 1967), Gaussian Mixture Models (GMM)(McLachlan & Peel, 2000), and density-based methods such as DBSCAN(Ester et al., 1996). These algorithms were chosen to represent centroid-based, probabilistic, and density-driven approaches to grouping.

To quantitatively assess clustering quality, we computed three standard metrics for each algorithm. **Silhouette Score**(Rousseeuw, 1987), which measures cohesion and separation of clusters; the **Calinski–Harabasz Score**(Calinski & Harabasz, 1974), which evaluates variance between and within clusters; and the **Davies–Bouldin Score**(Davies & Bouldin, 1979), which captures the average similarity between clusters. The results of these metrics are summarized in Table 2. The resulting cluster assignments were subsequently integrated into the lesion attribute analysis and report generation pipeline.

For each cluster obtained from the latent feature analysis, we computed descriptive lesion-level attributes to characterize the aggregated properties of the samples. Specifically, the following quantitative measures were extracted : Lesion size (calculated as the total area of the binary segmentation mask) , Class distribution (proportion of benign, malignant, and non-lesion cases within the cluster), Shape compactness (defined as $\frac{4\pi \times \text{Area}}{\text{Perimeter}^2}$, capturing the regularity of lesion boundaries), Edge sharpness (estimated from the image intensity gradient along the lesion boundary), Centroid position (relative location of the lesion centroid within the image frame). The attributes were calculated first class-wise for the dataset. Then these attributes were aggregated at the cluster level by finding clusters in the dataset using an unsupervised clustering algorithm to provide structured, quantitative summaries that supported downstream interpretability and report generation.

To transform structured outputs from segmentation, latent space clustering, and lesion attribute analysis into a clinically interpretable summary, we employed large language models (LLMs) as generative report engines. The analysis results for each experiment were formatted into a structured textual input, which was combined with a consistent prompt designed to guide the model toward producing concise, human-readable reports. To evaluate robustness across different generative backends, we tested multiple state-of-the-art LLMs under identical prompting conditions. Specifically, we used the o3 model, Claude Sonnet 5, GPT-5, and Gemini 2.5 Pro (06-05 release). Each model was provided with the same input prompt and structured results, and their generated reports were recorded for subsequent comparison. The complete prompt template and the raw outputs from each model are presented in the Appendix for transparency and reproducibility. This setup enabled a systematic assessment of LLMs in their ability to synthesize quantitative findings into coherent, clinically oriented narrative reports.

## 3    RESULTS AND DISCUSSION

Across experiments on the BUS-UCLM dataset, three key findings emerged. DeepLabv3(Chen et al., 2018a)achieved the best segmentation performance, showing the highest IoU and Dice scores. In the latent space analysis, the k-means clustering produced the most coherent groupings, reflected in the superior clustering metrics. Finally, the four LLMs generated clear and clinically plausible lesion summaries, confirming the feasibility of translating quantitative attributes into human-readable reports.

To establish a fair and comprehensive benchmark for segmentation performance, we conducted evaluations across six different architectures: RefineNet Lin et al. (2017), SegNet Badrinarayanan et al. (2017), AttUNet Oktay et al. (2018), SK-UNet(Li et al., 2019), UNet (Ronneberger et al., 2015), and DeepLabV3 (Chen et al., 2018a). These architectures represent a diverse set of design philosophies, ranging from classical encoder–decoder structures to advanced models incorporating attention mechanisms and selective kernel operations. Table 1 provides a detailed summary of the quantitative results obtained for each model, reported across multiple well-established metrics, including Intersection-over-Union (IoU), Dice coefficient, pixel-wise accuracy, precision, recall, and F1 score. Such a comprehensive set of evaluation metrics ensures that the comparison captures both region-level and boundary-level performance, offering deeper insights into the strengths and weaknesses of each model. To maintain fairness and consistency, all models were trained under identical experimental settings. Specifically, training was performed with cosine annealing learning rate scheduling to allow for smooth adaptation of the learning rate, combined with an early stopping strategy to prevent overfitting and reduce unnecessary computation. This controlled setup guarantees that the performance differences observed across models can be attributed to architectural variations rather than discrepancies in training protocols.

Among the evaluated models, **DeepLabV3 achieved the highest overall performance**, with an average IoU of 75.28% and a Dice coefficient of 81.35%. This model consistently outperformed others

across malignant and benign lesion segmentation, while also requiring fewer parameters (15.3M) compared to heavier architectures such as RefineNet(Lin et al., 2017) (118M). UNet(Ronneberger et al., 2015) and AttUNet(Oktay et al., 2018) provided competitive results, especially for benign lesions, whereas SegNet(Badrinarayanan et al., 2017) and RefineNet(Lin et al., 2017) lag behind in both the lesion-specific metrics and the overall scores. In addition to quantitative evaluation, qualitative results are illustrated in Figure 2, showing sample images, ground-truth masks, and predictions for each model. As evident, DeepLabV3(Chen et al., 2018a) produces sharper and more accurate boundaries around lesions, capturing finer structures compared to other methods. These visual outcomes corroborate the superior numerical performance reported in Table 1.

Table 1: Comparison of segmentation performance across models on the BUS-UCLM dataset.

| Model | IoU (%) | Dice (%) | Accuracy (%) | Precision (%) | Recall (%) | F1 Score (%) | Learning Rate | Scheduler | Epochs | Train Time | Params |
|---|---|---|---|---|---|---|---|---|---|---|---|
| RefineNet(Lin et al., 2017) | Avg: 60.04
BG: 98.27
Ma: 36.63
Be: 45.34 | Avg: 73.71
BG: 99.11
Ma: 64.66
Be: 57.36 | Avg: 63.90
BG: 99.76
Ma: 41.95
Be: 49.98 | Avg: 85.86
BG: 98.51
Ma: 76.33
Be: 82.75 | Avg: 63.90
BG: 99.76
Ma: 41.95
Be: 49.98 | Avg: 71.28
BG: 99.13
Ma: 52.95
Be: 61.77 | 1e-4 | CosineAnnealingLR
$T_{max}=100$
$eta_min=1e-6$
startLR:1e-4 | 60 | 2046 sec | 118.0M |
| SegNet(Badrinarayanan et al., 2017) | Avg: 61.50
BG: 98.59
Ma: 39.84
Be: 46.11 | Avg: 67.35
BG: 99.28
Ma: 55.03
Be: 47.76 | Avg: 66.99
BG: 99.71
Ma: 47.18
Be: 54.08 | Avg: 84.15
BG: 98.88
Ma: 76.73
Be: 76.84 | Avg: 66.99
BG: 99.71
Ma: 47.18
Be: 54.08 | Avg: 72.82
BG: 99.29
Ma: 56.39
Be: 62.76 | 1e-4 | CosineAnnealingLR
$T_{max}=100$
$eta_min=1e-6$
startLR:1e-4 | 58 | 1396 sec | 29.44M |
| AttUnet(Oktay et al., 2018) | Avg: 69.30
BG: 98.20
Ma: 51.20
Be: 56.10 | Avg: 68.28
BG: 99.36
Ma: 57.64
Be: 47.84 | Avg: 72.44
BG: 99.66
Ma: 59.67
Be: 58.00 | Avg: 81.51
BG: 99.09
Ma: 66.12
Be: 79.31 | Avg: 72.44
BG: 99.66
Ma: 59.67
Be: 58.00 | Avg: 75.70
BG: 99.37
Ma: 61.94
Be: 65.78 | 1e-4 | CosineAnnealingLR
$T_{max}=100$
$eta_min=1e-6$
startLR:1e-4 | 61 | 1883 sec | 31.397M |
| SKUNet(Li et al., 2019) | Avg: 65.98
BG: 98.75
Ma: 44.70
Be: 54.49 | Avg: 72.40
BG: 99.36
Ma: 62.68
Be: 55.17 | Avg: 71.29
BG: 99.79
Ma: 51.07
Be: 63.02 | Avg: 87.07
BG: 98.96
Ma: 79.90
Be: 82.34 | Avg: 71.29
BG: 99.79
Ma: 51.07
Be: 63.02 | Avg: 77.04
BG: 99.37
Ma: 61.49
Be: 70.25 | 1e-4 | CosineAnnealingLR
$T_{max}=100$
$eta_min=1e-6$
startLR:1e-4 | 42 | 2464 sec | 46.33M |
| UNet(Ronneberger et al., 2015) | Avg: 67.90
BG: 98.75
Ma: 48.18
Be: 56.78 | Avg: 72.92
BG: 99.36
Ma: 61.14
Be: 58.27 | Avg: 73.45
BG: 99.74
Ma: 57.87
Be: 62.74 | Avg: 86.83
BG: 99.00
Ma: 73.90
Be: 86.18 | Avg: 73.45
BG: 99.74
Ma: 57.87
Be: 62.74 | Avg: 78.66
BG: 99.37
Ma: 64.51
Be: 72.10 | 1e-4 | CosineAnnealingLR
$T_{max}=100$
$eta_min=1e-6$
startLR:1e-4 | 76 | 2146 sec | 31.390M |
| DeeplabV3(Chen et al., 2018a) | Avg: 75.28
BG: 99.74
Ma: 63.43
Be: 62.68 | Avg: 81.35
BG: 99.38
Ma: 78.17
Be: 66.51 | Avg: 75.28
BG: 99.74
Ma: 63.43
Be: 62.68 | Avg: 87.14
BG: 99.06
Ma: 76.64
Be: 85.74 | Avg: 75.28
BG: 99.74
Ma: 63.43
Be: 62.68 | Avg: 80.07
BG: 99.39
Ma: 69.12
Be: 71.68 | 1e-4 | CosineAnnealingLR
$T_{max}=100$
$eta_min=1e-6$
startLR:1e-4 | 24 | 2046 sec | 15.31M |

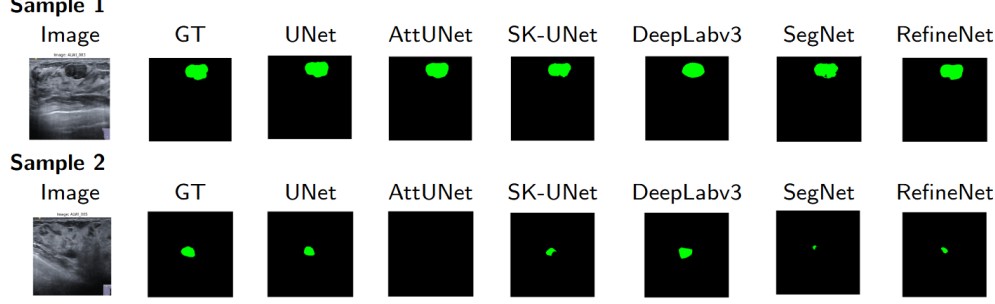

Figure 2: Qualitative segmentation results. From left to right: input ultrasound image, ground-truth mask, and predictions from UNet(Ronneberger et al., 2015) , AttUNet(Oktay et al., 2018) , SKUNet(Li et al., 2019) , DeepLabV3(Chen et al., 2018a) , SegNet(Badrinarayanan et al., 2017) , and RefineNet(Lin et al., 2017) . DeepLabV3 outputs exhibit clearer and more precise delineation of lesion regions.

To gain deeper insights into the behavior of the models during training, we additionally monitored the optimization dynamics throughout the entire training process. Figure 3 presents both the validation loss and training loss curves for a set of representative models, providing a clear view of how each model's performance evolved over time. As illustrated in the figure, all models exhibited stable convergence patterns, with minimal indications of overfitting, highlighting the robustness of the training process. This observation was further supported by the implementation of early stopping, which prevented unnecessary over-training, as well as the use of the cosine annealing learning rate scheduling strategy, which contributed to smoother and more efficient convergence. Among the different architectures that were evaluated, DeepLabV3 demonstrated a notably faster convergence

rate in comparison to the others, a trend that aligns well with its superior segmentation performance, as reported earlier in Table 1.

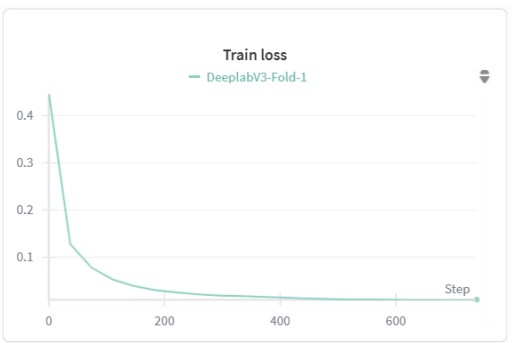 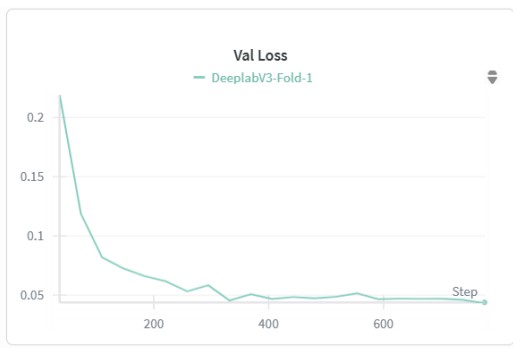

(a) Training loss curves across epochs.    (b) Validation loss curves across epochs.

Figure 3: Training dynamics of segmentation models tracked using. Both training and validation losses decrease smoothly, with convergence reached after early stopping. DeepLabV3 shows the most stable validation performance, consistent with its superior segmentation metrics.

## 3.1 Latent Feature Visualization and Clustering Results

To further investigate the internal feature representations captured by the segmentation network, we focused on encoder-level embeddings extracted from the best-performing model, DeepLabV3. These embeddings, which consist of high-dimensional feature vectors, were reduced to a two-dimensional space using t-SNE, enabling qualitative inspection of the latent structure. The visualization is presented in Figure 4, where clear grouping patterns emerge among the lesion samples. Notably, these clusters show loose correspondence to the underlying benign, malignant, and background classes, suggesting that the encoder is not only effective in driving accurate segmentation outcomes but also in learning semantically meaningful and discriminative latent representations. This observation highlights the dual role of the encoder in both pixel-level prediction and higher-level semantic organization of lesion features.

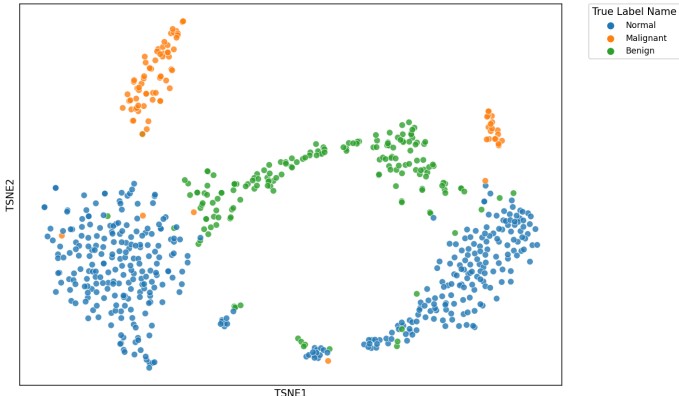

Figure 4: t-SNE projection of latent features extracted from DeepLabV3 encoder. Clear pre-existing clustering patterns emerge across lesion categories (Normal, Malignant, Benign), suggesting discriminative feature organization in the learned embedding space.

| Algorithm / Technique | Silhouette | Calinski–Harabasz | Davies–Bouldin |
|---|---|---|---|
| K-Means (MacQueen, 1967) | **0.6575** | **764.9680** | 0.8388 |
| Agglomerative Clustering (Johnson, 1967) | 0.6544 | 760.3402 | 1.1002 |
| Spectral Clustering (von Luxburg, 2007) | 0.5455 | 613.1636 | 1.0150 |
| Gaussian Mixture Model (GMM) (McLachlan & Peel, 2000) | 0.6566 | 764.9143 | **0.8345** |
| BIRCH (Zhang et al., 1996) | 0.6474 | 703.7319 | **0.5070** |
| DBSCAN (Ester et al., 1996) | -0.1502 | 47.8186 | 1.4776 |
| OPTICS (Ankerst et al., 1999) | -0.4360 | 16.9120 | 1.3065 |
| MeanShift (Comaniciu & Meer, 2002) | 0.5272 | 347.3730 | 0.6819 |
| Affinity Propagation (Frey & Dueck, 2007) | 0.3971 | 579.1452 | 0.8662 |

Table 2: Comparison of clustering algorithms applied to latent features extracted from DeepLabV3. Higher Silhouette and Calinski–Harabasz scores and lower Davies–Bouldin scores indicate better clustering quality.

To quantitatively evaluate latent structure, we applied a suite of clustering algorithms, including $k$-means MacQueen (1967), Agglomerative Clustering Johnson (1967), Spectral Clustering von Luxburg (2007), Gaussian Mixture Models (GMM) McLachlan & Peel (2000), BIRCH Zhang et al. (1996), DBSCAN Ester et al. (1996), OPTICS Ankerst et al. (1999), MeanShift Comaniciu & Meer (2002), and Affinity Propagation Frey & Dueck (2007). Clustering quality was assessed using the Silhouette Score Rousseeuw (1987), Calinski–Harabasz Index Calinski & Harabasz (1974), and Davies–Bouldin Score Davies & Bouldin (1979). The results are reported in Table 2. Among the evaluated methods, $k$-means achieved the strongest overall performance across metrics, followed closely by GMM. We therefore adopted the $k$-means clustering results for downstream lesion attribute analysis and report generation, as this method provided the clearest partitioning of lesion types in the latent space.

Beyond evaluating segmentation and clustering performance, we aimed to understand whether the learned latent space representations could be leveraged for clinically relevant lesion characterization. To this end, we calculated descriptive lesion-level metrics both for the ground-truth class labels (Normal, Benign, Malignant) and for the unsupervised clusters obtained via $k$-means (identified as the best-performing clustering method).

When computed on the basis of ground-truth labels, the metrics showed clear differences across lesion categories (Table 3). Normal samples exhibited trivial lesion attributes, reflecting the absence of true lesions. In contrast, benign lesions tended to be more compact with sharper edges, while malignant lesions displayed larger irregularities and higher sharpness, consistent with clinical expectations.

| Class | Samples | Avg. Size | Avg. Compactness | Avg. Sharpness | Class Distribution (%) |
|---|---|---|---|---|---|
| Normal | 419 | 65536.0 | 0.0 | 0.0 | 100-0-0 |
| Benign | 174 | 65230.7 | 0.97 | 22.32 | 0-100-0 |
| Malignant | 90 | 65128.7 | 0.79 | 18.87 | 0-0-100 |

Table 3: Lesion-level metrics computed per ground-truth class (Normal, Benign, Malignant). Values are averaged over samples in each category.

Next, we analyzed the lesions within the unsupervised clusters identified by $k$-means, as summarized in Table 4. The resulting clusters revealed a meaningful separation of lesion subtypes in the latent space, highlighting the capability of the clustering to capture underlying structural differences among samples. Specifically, Cluster 1 consisted primarily of normal samples, Cluster 2 contained a mixture of normal and benign lesions, and Cluster 3 was enriched for both malignant and benign lesions, reflecting a more complex composition. Notably, the lesion attributes, such as sharpness and compactness, exhibited patterns that were consistent with clinical expectations, effectively distinguishing malignant from benign classes. This alignment between the latent cluster structures and clinically relevant lesion characteristics underscores the interpretability and practical relevance of the unsupervised clustering approach.

| Cluster | Samples | Avg. Size | Avg. Compactness | Avg. Sharpness | Class Distribution (%) |
|---------|---------|-----------|------------------|----------------|------------------------|
| Cluster 1 | 369 | 65443.6 | 0.29 | 7.54 | 65.9-18.7-15.4 |
| Cluster 2 | 273 | 65429.1 | 0.32 | 7.71 | 64.5-30.0-5.5 |
| Cluster 3 | 41 | 64889.2 | 1.12 | 16.90 | 0-56.1-43.9 |

Table 4: Cluster-wise lesion metrics for $k$-means clusters computed from DeepLabV3 latent features. Class distribution percentages indicate the proportion of Normal-Benign-Malignant samples within each cluster.

These results confirm that unsupervised grouping of latent features can recover clinically interpretable lesion characteristics, even without explicit label supervision. In particular, Cluster 3 showed high compactness and sharpness with a strong enrichment of malignant lesions, underscoring the potential of latent representations for automated lesion profiling and risk stratification. Finally, the extracted lesion-level attributes (both per-class and cluster-based) were used as structured inputs for the subsequent large language model (LLM)–driven report generation stage.

Following the extraction of latent features and lesion-level metrics, we performed automated clinical report generation using multiple LLMs. The models included OpenAI GPT-5, O3, Claude Sonnet 5, and Gemini 2.5 Pro. Each model was provided with:

- Latent space visualization (pre-clustering) of lesions.
- K-means clustered latent space visualization.
- Cluster-wise lesion metrics (mean lesion size, compactness, sharpness, centroid position, class distribution).

The prompt guided the models to:

1. Generate a human-interpretable clinical report.
2. Explain what latent visualizations and clustering suggest about data distribution.
3. Compare clusters in terms of lesion metrics.
4. Suggest possible clinical interpretations, including BI-RADS context.
5. Highlight alignment with medical categories and detect potential novel subgroups.

Across the four models, the generated reports consistently conveyed the following insights:

- **Latent Feature Separability:** Pre-clustering visualizations demonstrated clear separation of normal, benign, and malignant lesions, indicating that the AI model captures clinically relevant patterns.
- **K-Means Clustering Interpretation:** Unsupervised clusters largely aligned with ground-truth labels, with Cluster 3 revealing a mixture of benign and malignant lesions, suggesting ambiguous or borderline cases.
- **Lesion Metrics Comparison:** Cluster-wise metrics consistently showed that benign lesions are highly compact with sharp boundaries, malignant lesions have slightly lower compactness with noticeable sharpness, and mixed clusters (K-means cluster 3) contained lesions with the highest compactness and sharpness, representing potential borderline cases.
- **Clinical Interpretation:** Reports correctly associated clusters with BI-RADS categories:
  - Normal → BI-RADS 1 (Negative)
  - Benign → BI-RADS 2-3 (Probably Benign)
  - Malignant → BI-RADS 4-5 (Suspicious or Highly Suggestive of Malignancy)
  - Mixed cluster → BI-RADS 3/4a, indicating ambiguous or borderline lesions requiring further evaluation.
- **Subgroup Discovery:** All models highlighted the existence of ambiguous lesions that could benefit from additional clinical scrutiny or follow-up, demonstrating the LLM's ability to contextualize clustering-derived insights.

- **Consistency Across Models:** While minor differences in phrasing and emphasis existed, the key findings and clinical interpretations were consistent across GPT-5, O3, Claude Sonnet 5, and Gemini 2.5 Pro.

The prompt successfully guided all models to produce structured, clinically meaningful reports. The latent visualizations and lesion metrics served as informative inputs, allowing LLMs to bridge AI-derived insights with human-interpretable interpretations. Detailed prompts and full model responses for each LLM are provided in Appendix A. This allows readers to examine individual outputs and the differences in wording, emphasis, and clinical interpretation. The combination of latent feature extraction, unsupervised clustering, and LLM-based report generation provides a robust pipeline for producing human-interpretable clinical analyses of breast ultrasound images. The methodology not only captures known lesion categories but also identifies ambiguous subgroups, potentially improving triage and BI-RADS-based clinical decision-making.

## 4 CONCLUSION

We have introduced a segmentation-driven latent probing framework that unifies high-performance lesion segmentation, unsupervised clustering, quantitative attribute analysis, and LLM-based clinical description generation without requiring paired image–text data. In contrast to prior BUS interpretability work, which relies on pixel-level saliency or large multimodal datasets, our approach surfaces structured, cluster-level semantics directly from model embeddings. This enables transparent and clinically aligned summaries that preserve the diagnostic patterns encoded in deep latent spaces while maintaining state-of-the-art segmentation performance. The proposed methodology is architecture and modality-agnostic, offering a scalable blueprint for bridging the semantic gap between deep visual features and domain-expert reasoning in any low-resource imaging context. By reframing explainability as the quantitative-to-language mapping of latent clusters, this work opens a new direction for unsupervised, interpretable, and multimodally integrated medical AI that extends well beyond breast ultrasound.

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

## A  PROMPT TEMPLATE AND LLM OUTPUTS

This appendix contains the complete prompt template used for guiding the large language models (LLMs) and links to the raw outputs (hosted as external PDFs).

### A.1  PROMPT TEMPLATE

Listing 1: Prompt template used for LLMs.

```
You are an expert in medical imaging and breast lesion analysis (BI-RADS
    ↪ context).
I have extracted latent features from an AI model trained on ultrasound
    ↪ images.
Here are the visualizations and metrics:
the images are in order.....
1. Latent space visualization (before clustering) -> shows the
    ↪ separability of lesions.
2. K-means clustered latent space visualization.
3. Cluster-wise lesion metrics (mean lesion size, shape irregularity,
    ↪ intensity variation, etc.): (for latent features cluster)

Cluster 1: Label: Normal  |  Samples: 419
{'average_lesion_size': 65536.0, 'average_compactness': 0.0,
    ↪ 'average_sharpness': 0.0,
 'average_centroid_position': array([0., 0.]),
    ↪ 'average_class_distribution':
 {'normal': 100.0, 'benign': 0.0, 'malignant': 0.0},
    ↪ 'average_confidence': None}

Cluster 2:Label: Benign  |  Samples: 174
{'average_lesion_size': 65230.69540229885, 'average_compactness':
    ↪ 0.9696500361999636,
 'average_sharpness': 22.32119994918475, 'average_centroid_position':
 array([0.32666498, 0.54290669]), 'average_class_distribution':
 {'normal': 0.0, 'benign': 100.0, 'malignant': 0.0},
    ↪ 'average_confidence': None}

Cluster 3: Label: Malignant  |  Samples: 90
{'average_lesion_size': 65128.65555555555, 'average_compactness':
    ↪ 0.7932100618747839,
 'average_sharpness': 18.86858363038208, 'average_centroid_position':
 array([0.39485242, 0.49573618]), 'average_class_distribution':
 {'normal': 0.0, 'benign': 0.0, 'malignant': 100.0},
    ↪ 'average_confidence': None}

4) Cluster wise lesion metrics for clusters made by K-means algorithm

Cluster 1: 369 samples
{'average_lesion_size': 65443.62059620596, 'average_compactness':
    ↪ 0.2932589859253656,
 'average_sharpness': 7.544118723283007, 'average_centroid_position':
 array([0.11817372, 0.18333291]),
 'average_class_distribution': {'normal': 65.85365853658537, 'benign':
    ↪ 18.69918699186992,
 'malignant': 15.447154471544716}, 'average_confidence': None}

Cluster 2: 273 samples
{'average_lesion_size': 65429.12454212454, 'average_compactness':
    ↪ 0.3150485932852659,
 'average_sharpness': 7.711378741167081, 'average_centroid_position':
 array([0.11716137, 0.18562938]),
 'average_class_distribution': {'normal': 64.46886446886447, 'benign':
    ↪ 30.036630036630036,
 'malignant': 5.4945054945054945}, 'average_confidence': None}
```

```
Cluster 3: 41 samples
{'average_lesion_size': 64889.19512195122, 'average_compactness':
    ↪ 1.1191995144923599,
 'average_sharpness': 16.904271040548537, 'average_centroid_position':
 array([0.40939679, 0.5062282 ]),
 'average_class_distribution': {'normal': 0.0, 'benign':
    ↪ 56.09756097560975,
 'malignant': 43.90243902439025}, 'average_confidence': None}

Tasks:
- Generate a Human interpretable clinical report for the analysis of the
    ↪ data
- Explain what the latent visualization and clustering suggest about the
    ↪ data distribution.
- Compare clusters in terms of lesion metrics.
- Suggest possible clinical interpretation (e.g., relation to benign vs
    ↪ malignant, BI-RADS categories).
- Highlight if clustering aligns with expected medical categories or
    ↪ reveals new subgroups.
```

## A.2 LLM Outputs (external PDFs)

The complete raw outputs for each model are provided as separate PDF files (hosted externally). Use the links below:

- GPT-5 Outputs (PDF)
- Claude Sonnet 5 Outputs (PDF)
- Gemini 2.5 Pro Outputs (PDF)
- o3 Outputs (PDF)

