# OpenReview forum: "Lesions, Latents, and Language: Interpreting Breast Ultrasound Features via Latent Probing and LLM-Driven Report Synthesis"
_ICLR.cc/2026/Conference — ICLR 2026 Conference Withdrawn Submission_

### Official Review · Reviewer_EW5V · 2025-10-15

**Soundness:** 3
**Presentation:** 3
**Contribution:** 2
**Rating:** 4
**Confidence:** 4

**Summary:**

This paper focus on segmentation of breast ultrasound lesions by extracting latent features. After getting these latent features, the authors run different clustering algorithms in the latent feature space and measure the qualities of these clustering methods by 3 different metrics (Silhouette score, Calinski–Harabasz score and Davies–Bouldin score). Then the authors decide to use k-means algorithm. After clustering these features, they extract cluster-level features which include lesion size, sharpness, class distribtuion. Along with the latent space visualization and clustered latent space visualization, all of these information is given to 4 different LLMs to extract report. These reports include information about data distribution, cluster differences/similarities, possible new subgroup/subcluster detection.

**Strengths:**

-Authors run segmentation experiments on a relatively new dataset. And they provide detailed results for all popular segmentation models.
-Clustering of latent features is also done in detail. The authors not only provide 9 different clustering methods, they also assess these methods by 3 different metrics for clustering.
-The idea of generating reports by using latent space information seems interesting. As the authors provided these latent space features may include further information than just the class of the lesion.

**Weaknesses:**

-It is unclear what these generated reports can be used for. Normally, reports per sample can be used for detection/diagnosis etc. However, the usage of cluster-level or dataset-level reports is unclear to me.
-Even if the very high level idea of generating reports from latent features seems interesting, when I look at the results in the paper, these are not pointing anything novel/interesting. For example, in Table 1, the message is that the authors tried different models on a relatively new dataset and pick the best model. The Figure 3 is also expected a little bit irrelevant. Similarly, even if  the Table 2 is a good table, so that we know why the k-means is chosen, it is also not related to the main story of the paper.

**Questions:**

-In the paper, we eventually get cluster-level clinical interpretation. How can we use these reports ? Can it be used to classify samples within the cluster in terms of BI-RADS scores ? I want to understand the usacase of these generated reports. Similarly, why are we clustering these samples in the first place ?
-You mention each LLM was provided with latent space visualization of lesions and K-means clustered space visualization. When I look at the generated reports I see these files : "1. tsne_True_Label_Name.jpg and 2. tsne_KMeans.jpg". So, are those two figures tsne plots which includes data samples with colors of true labels and clusters respectively. Are these figures similar to Figure 4 in the paper ?

---

### Official Review · Reviewer_RSSd · 2025-10-29

**Soundness:** 1
**Presentation:** 2
**Contribution:** 1
**Rating:** 0
**Confidence:** 3

**Summary:**

They trained several segmentation models on breast ultrasound images and chose one (DeepLabV3). Follow up experimentation was with the selected model, extracting its internal latent features, grouping them using several clustering algorithms and selecting the top performing (k-means clustering). They measured the quantitative properties of each cluster such as average lesion shape and sharpness.
They then fed these summaries to black-box "state-of-the-art" LLMs and prompted them to generate human-interpretable clinical reports.
Their main contribution is this pipeline itself which contains a "explainability interface" that bridges the gap between a segmentation model's output and a specialized clinical reasoning, all without needing any pre-existing text reports to train on.

**Strengths:**

The primary originality is the framework itself. Proposing a new way to generate decision explainability by using latent features clusters. It offers an explainability interface without the need for actual ground-truth reports from specialized personnel, a crucial bottleneck in training models using medical imaging (scarcity and size of datasets). The paper is clear and direct to the point.

**Weaknesses:**

In the discussion of the t-SNE plot (figure 4), they claim "clear grouping patterns emerge", and following up they admit that the clusters show "loose correspondence" to the underlying benign, malignant and background classes (also, it’s very confusing in the paper when mentioning "background" and "normal" classes). "Clear" and "loose" are contradictory, the visualization looks distinct but it may not be clinically meaningful.

Given that, the "best" k-means results (Table 4) proves this: "Cluster 3" may not be a "novel subgroup", as claimed, but a failed cluster, with a near 50/50 mix of "benign" (56.1%) and "malignant" (43.9%). The authors also ignored the "negative Silhouette Score" of DBSCAN (Table 2), which strongly suggests that the "clear grouping patterns" mentioned in the t-SNE plot (Figure 4) may just be an artifact of the visualization itself, not a true structure.

In Phase 3, the flaws were regarding the confirmation bias. The “Prompt Template” (Appendix A) explicitly told the models to “Suggest possible clinical interpretation (e.g... BI-RADS categories)” and "detect... new subgroups". The "discovery" that the models did this was not a sign of a specialized report, instead it seems it was just a "prompt-following" directive. Furthermore, the evaluation that the resulting reports were "clear and clinically plausible" was subjective, as they provide no blinded clinical expert scoring.


The main problem is that it is very dificult to validate the entire architecture because the discovered clusters would need to be "generalizable" to new instances during inference, otherwise the entire architecture is not usable in real world scenarious.

**Questions:**

Your interpretation of the latent space in Phase 2 is questionable. The "subgroup discovery" relies on interpreting "Cluster 3", which contains only 41 samples and is a 56%-44% mix of benign and malignant lesions. How can you justify that this is a "clinically interpretable" subgroup rather than a statistical artifact of k-means failing to partition a poorly-separated latent space? This concern is amplified by your own data in Table 2, which shows that density-based algorithms like DBSCAN and OPTICS failed with negative Silhouette Scores, a finding that is not addressed throughout the paper. This suggests that the clear grouping patterns in your t-SNE plot may be an artifact of the visualization rather than a true property of the latent space.

The core claim of emergent clinical reasoning in Phase 3 appears to be a product of confirmation bias. The "prompt template" in Appendix A explicitly instructs the LLMs to "suggest possible clinical interpretation (e.g... BI-RADS categories)" and "detect... new subgroups". Can you please explain how this demonstrates reasoning from the quantitative data, rather than simple prompt-following? A more convincing experiment would involve a more concise prompt ("Describe these clusters") to check if these clinical concepts emerge organically as an output.

The evaluation of the LLM-generated reports is entirely subjective. You claim the reports are "clear and clinically plausible" but provide no objective evidence. To substantiate your claim that you "close the semantic gap", the paper requires a rigorous evaluation. Can you provide any quantitative text-analysis metrics or, ideally, a blinded evaluation by qualified personnel scoring the reports for clarity, accuracy, and clinical utility? Without this, the paper's main conclusion is unsubstantiated.

---

### Official Review · Reviewer_iwfb · 2025-10-30

**Soundness:** 2
**Presentation:** 1
**Contribution:** 1
**Rating:** 0
**Confidence:** 4

**Summary:**

This paper addresses the lack of interpretability and semantic transparency in current medical imaging models, particularly in breast ultrasound lesion analysis, by introducing a segmentation-driven latent probing framework that connects deep visual features with natural-language reasoning without requiring paired image–text data. The resulting framework leverages LLMs to summarize the latent feature geometry and some heuristic-based lesion features and generate report.

**Strengths:**

1. The paper comprehensively compared the segmentation performance of various models on the BUS-UCLM dataset

**Weaknesses:**

1. The specific problem addressed by this paper is not well established. The authors claim to tackle the absence of “a unified framework that connects latent visual representations from top-performing segmentation models to clinically meaningful natural-language explanations without requiring paired image–text data.” However:
(a) The experiments do not utilize or benchmark against state-of-the-art segmentation models (e.g., Swin-UNETR, MedSAM), which weakens the claimed contribution.
(b) Numerous prior works have explored connecting latent representations to natural-language explanations (e.g., [1, 2]), suggesting that the proposed problem setting is not entirely novel.
(c) Although the paper claims to achieve natural-language explanations without image–text paired supervision, the approach implicitly depends on large language models that have already been trained on extensive multimodal data, thereby indirectly leveraging paired vision–language knowledge.
2. The paper does not introduce a substantial methodological novelty beyond combining existing segmentation, clustering algorithms, and language modeling components. The pipeline primarily integrates off-the-shelf modules without presenting new learning objectives, architectures, or theoretical insights. Consequently, the contribution appears incremental relative to the existing literature on interpretable vision–language modeling.

[1] Yang, Yu, Seungbae Kim, and Jungseock Joo. "Explaining deep convolutional neural networks via latent visual-semantic filter attention." Proceedings of the IEEE/CVF Conference on Computer Vision and Pattern Recognition. 2022.
[2] Schmalwasser, Laines, et al. "Exploiting text-image latent spaces for the description of visual concepts." International Conference on Pattern Recognition. Cham: Springer Nature Switzerland, 2024.

**Questions:**

There are no questions; I don't think this should be accepted.

---

### Official Review · Reviewer_Ugm3 · 2025-10-31

**Soundness:** 3
**Presentation:** 2
**Contribution:** 2
**Rating:** 2
**Confidence:** 4

**Summary:**

This manuscript presents a framework for bridging visual representations with natural language, for breast ultrasound lesions. A supervised segmentation model is first used to extract latent features for each lesion. These lesion features are then projected to lower-dimensional latent space and clustered. LLMs are then prompted with cluster-level summaries to produce natural language descriptions for the lesions.

**Strengths:**

- (Originality) Usage of LLMs to describe lesion appearance using latent features

 - (Quality) Extensive analysis of lesion segmentation models

**Weaknesses:**

- The study ultimately appears composed of three main steps: a lesion segmentation experiment involving six existing models, a manually-applied t-SNE clustering on extracted lesion features from the previous step, and finally an investigation of the ability of four LLMs to describe the data from the second step in human-readable form. It is unclear as to whether any of these steps represents a substantial contribution in their current form

 - Lack of quantitative analysis of the final human-reported report outputs, despite claims that "unsupervised clusters largely aligned with ground truth labels" and "Reports correctly associated clusters with BI-RADS categories etc"

 - While it is claimed that the generated structured reports are clinically meaningful, it is unclear whether they have true clinical application; this could have been achieved by evaluating the generated reports against actual ultrasound reports, and/or surveying end-user clinicians

**Questions:**

N/A

---

### Note · Authors · 2025-11-22

I have read and agree with the venue's withdrawal policy on behalf of myself and my co-authors.